# Recent Advancements in Hyperspectral Image Reconstruction from a Compressive Measurement

**DOI:** 10.3390/s25113286

**Published:** 2025-05-23

**Authors:** Xian-Hua Han, Jian Wang, Huiyan Jiang

**Affiliations:** 1Graduate School of Artificial Intelligence and Science, Rikkyo University, Tokyo 171-8501, Japan; 2School of Information Science and Engineering, ShanDong Normal University, Jinan 250358, China; jwang@sdnu.edu.cn; 3Software College, Northeastern University, Shenyang 110819, China; hyjiang@mail.neu.edu.cn

**Keywords:** hyperspectral image reconstruction, degardation, sensing mask, spatial–spectral modelling, long-dependency, MLP network

## Abstract

Hyperspectral (HS) image reconstruction has become a pivotal research area in computational imaging, facilitating the recovery of high-resolution spectral information from compressive snapshot measurements. With the rapid advancement of deep neural networks, reconstruction techniques have achieved significant improvements in both accuracy and computational efficiency, enabling more precise spectral recovery across a wide range of applications. This survey presents a comprehensive overview of recent progress in HS image reconstruction, systematically categorized into three main paradigms: traditional model-based methods, deep learning-based approaches, and hybrid frameworks that integrate data-driven priors with the mathematical modeling of the degradation process. We examine the foundational principles, strengths, and limitations of each category, with particular attention to developments such as sparsity and low-rank priors in model-based methods, the evolution from convolutional neural networks to Transformer architectures in learning-based approaches, and deep unfolding strategies in hybrid models. Furthermore, we review benchmark datasets, evaluation metrics, and prevailing challenges including spectral distortion, computational cost, and generalizability across diverse conditions. Finally, we outline potential research directions to address current limitations. This survey aims to provide a valuable reference for researchers and practitioners striving to advance the field of HS image reconstruction.

## 1. Introduction

Hyperspectral (HS) imaging extends beyond the conventional three-channel RGB representation by capturing a larger number of spectral bands across a continuous electromagnetic spectrum. This rich spectral resolution enables the differentiation of subtle material compositions and surface characteristics that remain indistinguishable in standard RGB imaging modalities. Consequently, HS images have been widely employed across diverse scientific and industrial domains, including image classification [1,2,3], remote sensing for land cover mapping and resource exploration [4,5,6,7], biomedical imaging for disease diagnosis and tissue characterization [8,9], and environmental surveillance for pollution detection and ecological monitoring. Traditionally, HS data acquisition relies on spectrometers that sequentially scan scenes along either the spectral or spatial dimensions, and thus inherently imposes significant temporal constraints. The sequential nature of such scanning methodologies renders them inefficient for capturing dynamically evolving phenomena, thereby limiting their applicability in real-time and high-speed imaging scenarios.

To address the limitations inherent in traditional HS imaging systems, snapshot compressive imaging (SCI) systems [10,11,12,13] have been developed, enabling the acquisition of HS images at video frame rates. Among these, Coded Aperture Snapshot Spectral Imaging (CASSI) [14,15,16] has emerged as a promising and innovative imaging modality that allows for the real-time capture of high-quality spatial and spectral information. CASSI leverages a coded aperture along with a disperser to modulate the HS signals across different wavelengths, effectively compressing them into a 2D measurement. Despite the promising potential of video-rate HS capture, the sensing capability of CASSI systems remains constrained by their limited ability to effectively capture sufficient incident light due to significant light attenuation in the coded aperture and dispersive elements. Moreover, the multiplexing nature of CASSI may also result in a low signal-to-noise ratio (SNR), particularly in low-light conditions or when capturing dynamic scenes. Consequently, enhancing the light efficiency of CASSI remains a critical challenge, necessitating the development of advanced optical designs, improved coding strategies, and noise-robust reconstruction algorithms to mitigate these constraints and optimize the system’s overall sensing performance. More recently, Motoki et al. [17] implemented a spatial–spectral encoding approach by integrating an array of 64 complementary metal oxide semiconductor (CMOS)-compatible Fabry–Pérot filters directly onto a monochromatic image sensor, dubbed as random array of Fabry–Pérot Filter (FAFP). This design enables efficient spectral differentiation while maintaining high optical transmission, thereby optimizing light throughput and minimizing signal loss. The system achieves a measured sensitivity of 45% for visible light, ensuring effective photon utilization, while the spatial resolution of 3 pixels at a 3 dB contrast level allows for accurate feature preservation. The integration of these filters facilitates high-fidelity hyperspectral image acquisition by reducing spectral cross-talk and enhancing the robustness of subsequent reconstruction algorithms. In addition, this optical encoding system operates at a frame rate of 32.3 frames per second (fps) at VGA resolution, meeting the practical requirements for real-time hyperspectral imaging applications. Similar to the CASSI system, the encoded measurement in the FAFP system remains a 2D snapshot image. Despite employing a different spectral encoding mechanism, FAFP shares the fundamental principle of compressing high-dimensional HS information into a lower-dimensional representation. This compression inherently introduces challenges in the reconstruction process, as recovering the full spectral–spatial information from a single 2D measurement remains an ill-posed inverse problem. Consequently, advanced computational reconstruction techniques are required to accurately retrieve the HS data while mitigating spectral distortion and preserving spatial details.

To tackle the challenging reconstruction task, various advanced methodologies have been proposed, including model-based optimization techniques, end-to-end (E2E) learning-based approaches, and deep unrolling pipelines. Traditional model-based methods [18,19,20,21,22] typically employ mathematical optimization frameworks that incorporate domain-specific priors, such as sparsity, low-rank constraints, or total variation regularization, to enhance reconstruction accuracy and mitigate ill-posedness. While these methods offer theoretical interpretability and robustness, they often suffer from high computational costs and limited adaptability to complex real-world scenarios. In contrast, E2E learning-based techniques [23,24,25,26,27,28,29,30] leverage deep neural networks to establish a direct mapping between the compressed measurements and the target HSIs, effectively bypassing the design of the hand-crafted priors. These methods have demonstrated remarkable performance improvements by learning inherent feature representations from large-scale datasets, enabling efficient and high-quality reconstructions. However, the black-box nature of deep learning models poses challenges in interpretability and generalization to unseen imaging conditions.

To bridge the gap between model-based and deep learning approaches, deep unrolling pipelines (DUN) [26,31,32,33,34] have emerged as a hybrid solution, integrating iterative optimization schemes with trainable neural network components. By unrolling traditional iterative algorithms into a finite number of network layers, these methods retain interpretability while potentially accelerating the reconstruction process through learned parameter adaptation. Moreover, the integration of data-driven prior learning through deep neural networks and the explicit model representation in DUNs has the potential to achieve high-fidelity HS image reconstruction.

Although these recent advancements have significantly improved the reconstruction of HS images from compressed measurements, several critical challenges remain unresolved, particularly in terms of computational efficiency and generalization in real-world imaging systems, necessitating further research efforts. The process of HS image reconstruction from highly compressed measurements continues to be a major bottleneck due to the inherent trade-offs between reconstruction accuracy, processing speed, and robustness under diverse imaging conditions. Therefore, a comprehensive review of existing reconstruction methods, along with a detailed analysis of their respective advantages and limitations, is essential to provide deeper insights into the current state of the field. Such an evaluation will not only facilitate a better understanding of the strengths and weaknesses of various approaches but also highlight key areas that require further innovation. Specifically, we begin by introducing the fundamental image formation models for HS images. We then provide a comprehensive review and comparison of three major reconstruction approaches: model-based methods, end-to-end learning-based techniques, and deep unfolding frameworks. Additionally, we briefly summarize the commonly used HS image datasets and performance evaluation metrics associated with these methods. To provide practical insights, we systematically evaluate and compare several representative algorithms through simulations on publicly available open-source datasets. Finally, we highlight key challenges and emerging trends in the field, aiming to inspire future research toward the development of more efficient, accurate, and robust HS image reconstruction frameworks.

**Distinction from Existing Surveys:** While several recent surveys [35,36] have reviewed HS image reconstruction techniques from RGB inputs captured by standard color cameras, the scope and focus of our study are fundamentally different. The surveys in [35,36] primarily address the problem of inferring HS data from RGB images, which is an under-constrained inverse problem relying heavily on learned spectral priors from external datasets. In contrast, our survey focuses on HS image reconstruction from compressive snapshot measurements acquired directly through specialized hyperspectral imaging systems such as CASSI [14,15,16] and FPFA [17]. This line of research involves different acquisition models, sensing hardware, and algorithmic formulations, often leveraging the physics of the measurement process. Our work provides a comprehensive categorization and analysis of model-based, deep learning-based, and hybrid reconstruction strategies specifically designed for compressive hyperspectral imaging. To our knowledge, there is currently no other survey that systematically reviews and contrasts these three paradigms within the context of compressive HS reconstruction.

## 2. Spectral Snapshot Imaging Model

The acquisition of a three-dimensional HS data cube comprising two spatial dimensions and one spectral dimension without employing scanning mechanisms remains a fundamental challenge. This limitation primarily stems from the constraints of conventional detector technologies, which are typically restricted to one-dimensional (1D) or two-dimensional (2D) sensor arrays. These conventional configurations are inherently incapable of capturing the complete spectral–spatial information simultaneously, necessitating alternative approaches to enable snapshot acquisition. Furthermore, the HS imaging process is intrinsically subject to trade-offs among several critical system parameters, including spectral resolution, spatial resolution, acquisition time, and optical throughput. Optimizing any one of these parameters often necessitates concessions in the others, making the design of high-performance systems a complex and tightly coupled optimization problem.

To address the above-mentioned limitations and facilitate real-time, compact, and efficient HS imaging, compressive sensing (CS) [37] frameworks have been widely adopted. CS leverages the inherent sparsity or compressibility of natural scenes in appropriate transform domains to enable the reconstruction of high-dimensional data from a significantly reduced number of measurements. By integrating signal sparsity assumptions with innovative optical encoding strategies, CS-based methods offer a viable pathway toward snapshot HS imaging with reduced hardware complexity and faster acquisition speeds. In recent years, a variety of compressive spectral imaging modalities have been developed and demonstrated significant advancements in dynamic spectral data acquisition. Notably, the Computed Tomography Imaging Spectrometer (CTIS) [38,39] employs a diffractive optical element to project the three-dimensional spectral data cube into a series of two-dimensional projections, which are subsequently reconstructed through tomographic algorithms. Similarly, the Coded Aperture Snapshot Spectral Imager (CASSI) [11,12,14] utilizes a coded aperture in conjunction with a dispersive element to modulate and shear the incoming light, enabling the encoding of spectral–spatial information into a single 2D snapshot. More recently, a novel HS sensor based on a CMOS-compatible random array of Fabry–Pérot Filters (FPFA sensor) [17] has been explored, and manifests great advantages of high sensitivity and compact size (CMOS compatible). To this end, these systems have collectively demonstrated remarkable potential for real-time and resource-efficient HS imaging. Next, we present the imaging process in the CASSI and the FPFA systems and their mathematical formulation.

**CASSI imaging system:** the CASSI imaging or encoding system primarily comprises the following key components: (1) imaging optics for capturing the light from the target scene; (2) coded aperture (sensing mask) as a spatial light modulator for encoding the incoming light in a spatially varying manner; (3) dispersive element for performing a wavelength-dependent shift to effectively spread the spectral components across the spatial domain; (4) a 2D focal plane array as a detector for integrating the spectrally shifted and spatially modulated light into a single 2D compressed measurement. An overview of the fundamental CASSI imaging process is illustrated in Figure 1. Specifically, let us consider a 3D HS image denoted by $X\in {ℜ}^{H\times W\times \Lambda }$, where *H* and *W* correspond to the spatial dimensions: height and width of the scene and Λ denotes the number of discrete spectral bands. In the CASSI system, a coded aperture (also referred to as a spectral sensing mask), represented as $M\in {ℜ}^{H\times W}$, is employed to spatially modulate the image at each spectral band λ. As a result, the modulated intermediate image for each spectral slice is formulated as:(1)$$Y_{\lambda }^{\prime }=X_{\lambda }⊙M,$$
where $X_{\lambda }$ denotes the 2D image corresponding to the λ-th spectral band, ⊙ indicates the element-wise multiplication, and $Y^{\prime }\in {ℜ}^{H\times W\times \Lambda }$ represents the modulated cube. Subsequently, the modulated spectral data cube $Y^{\prime }$ undergoes a wavelength-dependent spatial displacement along the horizontal axis, implemented by a dispersive optical element. This process introduces a spectral shear, whereby each spectral band is shifted laterally by an amount proportional to its wavelength index. Let *d* denote the dispersion step size, i.e., the number of pixels by which each successive spectral band is shifted. As a result, the spectrally dispersed data cube $Y^{\prime \prime }\in {ℜ}^{H\times (W+d(\Lambda -1))\times \Lambda }$ is generated, where the spatial width is extended to accommodate the cumulative shifts across all Λ spectral bands. The dispersive process can be formally described as follows:(2)$$Y_{\lambda }^{\prime \prime }\left(w,h\right)=Y_{\lambda }^{\prime }\left(w,h+d\left(\lambda \right)\right),$$
where *h* and *w* denote the spatial coordinates corresponding to the vertical and horizontal positions, respectively, in the imaging plane. After spectral dispersion, the resulting data cube $Y^{\prime \prime }$ contains spatially modulated and laterally shifted spectral components. To obtain the final 2D compressive measurement, denoted as Y, the system performs an integration along the spectral dimension. This spectral projection aggregates the energy contributions from all shifted spectral bands into a single 2D image on the detector plane. Mathematically, this process can be formulated as:(3)$$Y=\sum_{\lambda =1}^{\Lambda }Y_{\lambda }^{\prime \prime }+N,$$
where N represents the additive measurement noise, which arises from various sources such as sensor readout errors, photon shot noise, and thermal fluctuations during the acquisition process. To facilitate mathematical analysis and algorithmic implementation, the measurement model described from Equations (Equation 1)–(Equation 3) is reformulated using a matrix-vector representation. This compact form allows the compressive sensing problem to be addressed using linear algebraic tools and optimization frameworks. The reformulated model is expressed as follows:(4)y=Φx+n,
where Φ refers to the re-shaped sensing matrix, typically characterized by its fat (i.e., underdetermined) and highly structured nature. In most CASSI frameworks, the same coded aperture (sensing mask) is uniformly applied across all spectral bands, and then a dispersive element is incorporated to introduce a wavelength-dependent spatial shift, thereby enabling the spectral encoding of the 3D hyperspectral data cube into a 2D measurement through spatial integration.

**The FPFA sensor:** a recent approach [17] introduces a fundamentally different compressive imaging strategy based on spatial–spectral coded optical filters. This imaging method employs a random array of Fabry–Pérot filters monolithically integrated onto a CMOS sensor. Each filter cell transmits light at specific wavelengths, creating a spatially random spectral measurement matrix, where light passes through the filter array directly, with no dispersive element [17,40]. The spectral encoding is achieved via the transmittance patterns of the filters, which are designed to minimize correlation between wavelengths. Given a 3D HS image X and the equipped Fabry–Pérot filters $M\in {ℜ}^{H\times W\times \Lambda }$ with wavelength-dependent and spatially varying transmittance (weights), the imaging process in the FPFA sensor can be typically modeled by taking a weighted sum along the spectral dimension as:(5)$$Y=\sum_{\lambda =1}^{\Lambda }X\left(w,h,\lambda \right)⊙M\left(w,h,\lambda \right).$$

Similar to in the CASSI system, Equation (Equation 5) can be rewritten in a compact form: y=Φx+n, where Φ is typically sparse and highly structured, depending on the specific layout and transmission profile of the FP filters. It reflects the mapping from the 3D spectral space to the 2D measurement space. Thanks to the compact integration of the FPFA with CMOS sensors, the system benefits from the following: (1) high sensitivity, due to narrowband transmission of the resonant cavities; (2) compact and CMOS-compatible hardware, suitable for low-power and portable platforms; (3) single-shot and real-time operation, enabling applications in real-world consumer electronics such as smartphones and drones. The comparisons between the CASSI and FPFA sensors are illustrated in Table 1.

Following the spectral compressive measurement, a critical step involves reconstructing the latent 3D HS image x from the compressed 2D observation y. This inverse problem is inherently ill-posed due to the significant dimensionality reduction during acquisition. Section 3 will present the existing computational reconstruction methods for the spectral snapshot imaging systems.

## 3. Computational Reconstruction Methods

To date, a wide range of computational reconstruction techniques have been developed to address the challenges inherent in Coded Aperture Snapshot Spectral Imaging (CASSI) systems. These approaches primarily fall into three major categories: traditional model-based methods [11,18,19,20,21,22,41], deep learning-based methods [23,24,25,26,42,43] and the deep unfolding framework [31,33,34,44,45,46,47,48]. The taxonomy of the HS image reconstruction algorithms to be introduced is shown in Figure 2.

### 3.1. Model-Based Computational Reconstruction Methods

Model-based methods [11,18,19,20,21,22,41] formulate the HS image reconstruction as an inverse problem, leveraging explicit physical models of the spectral snapshot acquisition process. These methods typically incorporate domain knowledge through regularization terms such as sparsity, low-rank priors, or total variation, and solve the resulting optimization problem using iterative algorithms. Specifically, given the known sensing matrix Φ, a straightforward and intuitive strategy for estimating the latent HS signal x from the compressed measurement y is to minimize the reconstruction error via a data fidelity-driven objective. This heuristic approach can be formulated as the following optimization problem: (6)$$\hat{x}=\underset{x}{\arg\;\min}\frac{1}{2}{∥y-\Phi x∥}_{2}^{2},$$
where ${∥\cdot ∥}_{2}^{2}$ stands for the squared Euclidean norm. This formulation seeks the solution that best explains the observed measurement in the least-squares sense. It is well recognized that the HS image reconstruction problem from compressive measurements is severely ill-posed, as the number of unknown variables in the latent signal x far exceeds the number of observations in y. This imbalance results in an underdetermined linear system, making the accurate and robust recovery of the HS image extremely challenging.

To mitigate this issue, a substantial body of work has focused on incorporating hand-crafted image priors that capture the intrinsic structure and redundancies within HS data. These priors such as sparsity, low-rankness, smoothness, or spectral correlation serve to constrain the solution space and guide the reconstruction process toward more plausible results. Accordingly, the inverse problem is reformulated as a regularized optimization task, expressed as: (7)$$\hat{x}=\underset{x}{\arg\;\min}\frac{1}{2}{∥y-\Phi x∥}_{2}^{2}+\eta R\left(x\right),$$
where ${∥y-\Phi x∥}_{2}^{2}$ is the data fidelity term ensuring consistency with the measurement, R(x) denotes the regularization function encoding prior knowledge while η is a hyperparameter that balances the trade-off between measurement fidelity and prior enforcement. Most existing model-based approaches have focused on designing effective prior models and developing efficient optimization algorithms to enhance the stability, robustness, and accuracy of HS image reconstruction. In the following, we introduce several widely adopted and representative model-based reconstruction techniques based on the adopted priors that have demonstrated promising performance in the context of compressive HS imaging.

**Sparsity-based Prior:** Despite the inherently high ambient dimensionality of HS images, their intrinsic information content can often be effectively captured using only a limited number of active components when represented in a suitable transform or dictionary domain. This property reflects the inherent sparsity of HS data, which serves as a powerful prior to addressing the challenges of reconstruction from compressed measurements. Zhang et al. [49] presents a reconstruction method tailored for miniature spectrometers and combines sparse optimization with dictionary learning to effectively capture and represent the spectral signatures of the scene. By learning an over-complete dictionary from training data, the method promotes a sparse representation of the hyperspectral signal, enabling accurate reconstruction from a limited number of measurements. Figueiredo et al. [22] introduce an efficient gradient projection algorithm designed to solve sparse reconstruction problems via $L_{1}$-norm minimization. Operating within the compressed sensing framework, the method iteratively projects the solution onto a set defined by sparsity constraints, thereby efficiently solving the underdetermined inverse problem. Its generality and computational efficiency have made it a foundational algorithm, supporting a range of applications including hyperspectral image reconstruction where robust recovery from compressed measurements is critical. In addition, Wang et al. [19] exploits the nonlocal self-similarity of hyperspectral images by adaptively learning sparse representations from similar patches. By integrating nonlocal priors into the sparse reconstruction framework, the method effectively captures both spatial and spectral correlations, leading to enhanced restoration quality, especially in preserving fine details and reducing noise. Bioucas-Dias et al. [50] introduced the two-step iterative shrinkage/thresholding algorithm (TwIST) for solving the optimization task with multiple nonquadratic convex regularizers including sparsity. TwIST has popularly applied for the HS image reconstruction as a baseline model-based method.

**Total Variation (TV)-based Prior:** TV-based methods represent a prominent class of model-based approaches for HS image reconstruction. These methods are grounded in the assumption that natural images exhibit spatial piecewise smoothness, where intensity variations are often sparse and localized. In the context of HS images, this property is commonly exploited to preserve salient edges and suppress noise during the recovery process. A representative implementation is the Generalized Alternating Projection with TV(GAP-TV) [21], which alternates between enforcing data fidelity through projection onto the measurement constraint and applying TV denoising. Concretely, the GAP optimization iteratively updates the HS estimation by projecting the current estimate onto two sets: (1) data consistency set for enforcing y≈Φx and (2) prior constraint set for enforcing the TV regularization. Moreover, Eason et al. [51] proposed to solve the TV-regularized optimization problem for the CS inverse task with a continuation method, which involves gradually reducing the regularization parameter (or using a sequence of progressively “tighter” TV constraints). Such a scheme allows the optimization to start from an easier (more regularized) problem and slowly transition to the target problem, thereby improving both convergence speed and reconstruction quality. The method exhibits robustness to noise and measurement artifacts, making it well suited for practical compressive HS imaging applications.

**Low-Rank-based Prior:** By representing an HS image as a matrix or a tensor, one leverages the observation that the data often reside in a subspace of considerably lower dimension than the full ambient space. In other words, while the ambient space of an HS image with dimensions H×W×Λ is very high-dimensional, the intrinsic information contained in the data exhibits strong correlations among spectral bands (and, often, across spatial locations). Thus, the rank of the matrix representation or the multilinear rank of the tensor is typically much lower than the maximum possible rank. Inspired by the low-rank insight in HS imaging, Liu et al. [18] concentrate on global matrix rank minimization, employing a nuclear norm surrogate to enforce the low-dimensional structure of the underlying image. This approach demonstrates that by exploiting this global low-rankness, even when the number of measurements is significantly limited, it is possible to achieve high-quality reconstructions in snapshot compressive imaging systems. In contrast, Fu et al. [52] incorporate the low-rank property at a more localized level, taking advantage of the fact that HS images exhibit strong correlations not only across spectral bands but also among spatially similar regions. This dual exploitation of spectral and spatial redundancy leads to an enhanced restoration performance by better preserving fine details and reducing artifacts. Furthermore, Zhang et al. [41] extend the concept by treating HS data as a multidimensional tensor and applying dimension-discriminative low-rank tensor recovery. By directly modeling the inherent multi-way structure of HS images, the method retains the intrinsic spectral and spatial correlations that are often lost in vectorized formulations, culminating in superior reconstruction quality and robustness.

**Nonlocal-Self Similarity-based prior**: Nonlocal self-similarity (NSS) describes the intrinsic property of natural images wherein small, localized regions, referred to as patches or blocks, exhibit substantial resemblance to other spatially distant patches within the same image. This phenomenon is particularly pronounced in HS images, which, due to their high spectral resolution and inherent spatial redundancy, often contain repetitive spatial structures such as textures, edges, and homogeneous regions across different locations. By leveraging the NSS property, it becomes possible to more effectively constrain the ill-posed inverse problem of HS image reconstruction. This is achieved by enforcing that groups of nonlocally similar patches maintain a coherent, typically low-dimensional, structural representation. He et al. [20] exemplify the application of this concept by proposing an iterative framework that integrates nonlocal spatial denoising, guided by NSS priors, with a global low-rank spectral subspace model. While the primary objective of their work is HS image restoration (specifically denoising and inpainting), the underlying methodology grounded in NSS can be naturally extended to the context of compressive spectral snapshot reconstruction. Moreover, the NSS prior is conceptually aligned with low-rank modeling approaches, as both capitalize on structural redundancy. The integration of NSS-based regularization within HS reconstruction frameworks is further explored and validated in works such as [41,52].

Table 2 provides a comprehensive comparison of four representative categories of model-based HS image reconstruction methods. While these approaches effectively integrate knowledge of the physical image degradation process, leading to enhanced interpretability and well-defined mathematical formulations, they often face inherent limitations in representational capacity. This shortcoming primarily arises from the difficulty in designing hand-crafted priors that are both expressive and sufficiently generalizable across diverse scenes and imaging conditions. As a result, the performance of model-based methods may degrade when confronted with complex or nonstationary structures that deviate from the assumed prior assumptions.

### 3.2. Deep Learning-Based Reconstruction Methods

Motivated by the significant breakthroughs achieved by deep learning in various computer vision tasks, researchers have increasingly turned to data-driven approaches for HS image reconstruction. Among these, convolutional neural networks (CNNs) [24,25,27,53] and, more recently, Transformer-based architectures [29,42,43] have emerged as dominant frameworks due to their superior capacity to model complex spatial–spectral dependencies. These methods depart from traditional model-based paradigms by employing end-to-end (E2E) learning schemes, wherein a neural network is trained to directly learn a non-linear mapping from the compressed spectral measurements to the corresponding high-dimensional HS image [24,25,30,42,53].

This deep learning strategy eliminates the need for explicit prior modeling and physical forward operators, instead relying on large amounts of measurement/target pairs to implicitly learn underlying structures. Specifically, given a set of *N* training samples consisting of snapshot measurements and their corresponding HS targets, denoted as $\left(y_{n},x_{n}\right)$ for n=1,2,…,N, end-to-end (E2E) methods aim to learn a mapping function parameterized by θ that models the relationship between the snapshot inputs and the HS targets. This is achieved by minimizing a predefined loss function that quantifies the discrepancy between the reconstructed outputs and the ground-truth targets. When the loss is defined by the mean squared error (MSE), the optimization problem is given by:(8)$$\hat{\theta }=\underset{\theta }{\arg\;\min}\sum_{n=1}^{N}{∥x_{n}-F_{NN}\left(y_{n},\theta \right)∥}_{2}^{2},$$
where $F_{NN}\left(y_{n},\theta \right)$ denotes the output of the neural network parameterized by θ, taking the snapshot measurement $y_{n}$ as input and producing the corresponding HS reconstruction. Notably, conventional E2E learning methods [23,24,25,27,28,30,43,53,54] do not require prior knowledge of the sensing mask, as the reconstruction model implicitly learns the inverse mapping from the measurement space to the HS domain directly from data. Recently, several studies [29,42,55,56] have proposed incorporating the sensing mask into the network architecture to improve reconstruction performance. This paradigm has been explored in both CASSI systems [29,42] and FPFA-based sensors [55,56,57], where the integration of sensing priors into the learning process has demonstrated notable performance gains. In addition to the learning paradigm, the design of network architectures within E2E methods has been a critical factor influencing the performance of HS image reconstruction. The architecture directly determines the model’s capacity to capture and exploit the complex spatial–spectral correlations inherent in hyperspectral data. To this end, recent research efforts have focused on developing more effective and computationally efficient neural network architectures tailored for HS image reconstruction. These advancements have evolved progressively from CNNs [23,24,25], which are adept at capturing local spatial features, to Transformers [29,42,43], which leverage self-attention mechanisms to model long-range dependencies across spatial and spectral dimensions, to multi-layer perceptron (MLP)-based models [56], which offer enhanced modeling flexibility and global feature extraction through linear projections, and most recently to state space models such as Mamba [57], which provide efficient sequence modeling by combining long-range context capture with linear computational complexity. Subsequently, we present the representative architecture design in the E2E learning methods.

**CNN-based architecture:** CNNs have been widely adopted in early E2E HS image reconstruction methods due to their strong ability to capture local spatial structures and spectral correlations. These architectures typically consist of stacked convolutional layers that progressively extract hierarchical features from the input measurements. The use of small receptive fields enables the model to focus on fine-grained spatial patterns, while deeper layers can aggregate contextual information to improve reconstruction quality. For example, Hu et al. [24] investigate a high-resolution dual-domain learning strategy by utilizing a CNN architecture that jointly operates in both the spatial and spectral domains. Through dedicated convolutional blocks and multi-scale feature extraction, HDNet [24] effectively preserves spatial details while maintaining spectral fidelity. This dual-domain approach mitigates the common trade-off between resolution and spectral accuracy encountered in compressive spectral imaging. Miao et al. [25] propose a dual-stage generative model designed for efficient and high-quality HS image reconstruction, dubbed as λ-net. λ-net firstly employs a self-attention generator with a hierarchical channel reconstruction (HCR) strategy to generate an initial reconstruction of the HS image from the 2D snapshot measurement, and then a refinement stage composed of a small U-net and residual learning is developed to boost up the quality of reconstructed images from the first stage. Wang et al. [26] focuses on the incorporation of a learned deep spatial–spectral prior to the reconstruction process. The CNN architecture is purposefully designed to capture both local spatial context and long-range spectral dependencies. By integrating spatial and spectral cues into the feature extraction process, the network can resolve complex variations in hyperspectral data, leading to more faithful reconstructions. Meng et al. [23] embed spatial–spectral self-attention mechanisms into the CNN framework to explore feature dependencies more effectively. The self-attention modules enable the network to dynamically focus on the most informative spatial and spectral features, thereby enhancing the reconstruction accuracy. Moreover, the design prioritizes computational and hardware efficiency, making it particularly attractive for low-cost, end-to-end compressive spectral imaging systems. Moreover, Zhang et al. [28] emphasize both reconstruction quality and interpretability by integrating an optimal sampling strategy into its end-to-end learning process, thereby directly aligning the data acquisition with the reconstruction paradigm, while Cai et al. [58] introduce a binarization strategy into the CNN-based architecture. By employing binary weights and activations, the method achieves substantial reductions in memory usage and inference time, which is crucial for real-time or resource-constrained applications. More recently, Han et al. [30] employ Neural Architecture Search (NAS) to automatically discover optimal architectural configurations composed of diverse convolutional variants. The resulting architecture demonstrates both high efficiency and strong reconstruction performance, thereby validating the effectiveness of NAS in identifying specialized network structures tailored for HS image reconstruction.

Each aforementioned method leverages CNN-based architectures but distinguishes itself through unique design choices, ranging from optimization unrolling and dual-domain processing to architecture search and efficient quantization, to address specific aspects of the reconstruction problem. Despite their effectiveness, CNNs are inherently limited in capturing long-range dependencies due to the locality of convolutional operations, which has motivated the exploration of alternative architectures such as MLPs and Transformers in recent studies.

**Transformer-based architecture:** Transformer-based architectures have recently emerged as a powerful alternative to convolutional-based models in the E2E learning paradigm for HS image reconstruction. Originally developed for natural language processing and later adapted for vision tasks, Transformers are particularly well suited for modeling long-range dependencies across both spatial and spectral dimensions, a crucial aspect in HS imaging, where complex correlations often exist over distant regions in both domains. In the context of spectral snapshot measurements, Transformer-based models typically begin by exploring self-attention in the window-based spatial and spectral domain to reduce computatioal complexity, enabling the network to adaptively weigh the relevance of different spatial–spectral features [29,42,43,55,59,60,61]. Cai et al. [42] initially introduced a Transformer to HS image reconstruction with the main innovations: spectral-wise multi-head self-attention (S-MSA) and mask-guided mechanism (MM), dubbed as mask-guided spectral-wise Transformer (MST). The overall architecture follows popular encoder–decoder designs (U-Net) to progressively extract and fuse hierarchical features while keeping computational costs low. The encoder–decoder arrangement facilitates both local (via convolutions prior to attention) and global (via S-MSA) feature extraction. MST model achieves state-of-the-art reconstruction quality with lower memory and computational overhead by tailoring the attention mechanism to the spectral dimension and by incorporating prior physical knowledge (i.e., the mask). Takabe et al. [29] propose a versatile HS reconstruction framework that integrates a U-Net backbone with a spectral attention-based Transformer module. To enhance the model’s generalization capability across different measurement conditions, the training process incorporates a diverse set of sensing masks, enabling robust reconstruction performance under varying mask configurations. Cai et al. [43] introduce a coarse-to-fine sparse Transformer (CST) that leverages a multi-stage refinement strategy by integrating sparse attention mechanisms with hierarchical feature processing. This design effectively balances the modeling of global contextual information, critical for maintaining spectral fidelity, with the refinement of local spatial details, thereby achieving impressive HS image reconstruction. More recently, Wang et al. [59] investigate a spatial–spectral Transformer (denoted as S^2^-Tran) by using two interconnected streams with spatial and spectral self-attention modeling, respectively. Furthermore, inspired by the spatial nonuniform degradation (sensing mask), the S^2^-Tran directly integrates the mask information into both the network’s attention mechanism and its loss function to disentangle the entangled measurement caused by spatial shifting and mask coding. Luo et al. [60] propose a dual-window multiscale Transformer (DWMT) to capture both fine local details and broad contextual information by employing dual-window processing. Since DWMT realizes the attention mechanism separately within each window, reducing the computational burden compared to a global self-attention while still providing a sufficient receptive field to capture the complex spatial–spectral relationships. In addition, Yao et al. [55] propose a spatial–spectral cumulative-attention Transformer (SPECAT) for high-resolution HS image reconstruction in the FPFA sensor. The model leverages a dual-branch architecture to independently capture spatial and spectral dependencies through cumulative attention modules, enabling effective cross-dimensional feature fusion. By progressively integrating multi-scale context, SPECAT attains superior reconstruction quality while ensuring computational efficiency.

Overall, Transformer-based architectures have demonstrated state-of-the-art performance in HS image reconstruction, offering a powerful framework for extracting complex spatial–spectral features, particularly under compressed sensing scenarios such as snapshot spectral imaging.

**MLP-based architecture:** Although recent Transformer-based architectures have significantly advanced the efficiency of HS image reconstruction, their computational demands remain a critical bottleneck, particularly for deployment on real-world imaging systems with constrained resources. In response, emerging research has explored the substitution of the self-attention mechanism in Transformers with multi-layer perceptron (MLP)-like operations as a promising alternative [62,63,64,65]. These MLP-based architectures are adept at capturing non-local correlations and modeling long-range dependencies, offering competitive performance in various computer vision tasks while markedly reducing memory consumption and computational complexity. For instance, Cai et al. [66] construct a teacher-student network configured with Spectral–Spatial MLP (SSMLP) block for strategically fusing information from conventional RGB imaging and CASSI. This MLP-based model is designed to learn a robust mapping from the jointly encoded RGB–CASSI measurements to the corresponding HS images, and thus not only compensates for the reduced spectral resolution inherent to the CASSI data but also leverages the rich spatial details from the RGB inputs, thereby enhancing the overall reconstruction fidelity. Han et al. [56] employ cycle fully connected layers (CycleFC) to realize the spatial/spectral MLP-based model (S2MLP) for HS image reconstruction in the FPFA sensor, and further embed the mask information into the S2MLP learning branch to mitigate information entangling. Moreover, Cai et al. [67] extend the MLP paradigm by exploiting an adaptive mask strategy within a dual-camera snapshot framework, named as MLP-AMDC model. In this design, the dual-camera setup enables the simultaneous capture of complementary information streams, while the adaptive mask dynamically modulates the measurement process to optimize the acquisition of spectral data. The MLP-AMDC incorporates these adaptive measurements directly into its learning pipeline, using fully connected layers to integrate information from both cameras. With the explicit cues regarding the spatial and spectral encoding imposed during data collection, more accurate and robust recovery of HS images has been achieved from highly compressed snapshot measurements.

Overall, MLP-based architectures often achieve performance comparable to Transformer-based methods while being significantly more lightweight, positioning them as an attractive choice for deployment in real-world HS imaging systems.

**Mamba-based architecture:** Recent advances in sequence modeling have introduced structured state space models (SSMs), such as Mamba, as promising alternatives to attention-based networks in vision tasks. Leveraging the linear-time complexity and long-range dependency modeling capabilities of SSMs, Sp3ctralMamba [57] is the first method to apply a Mamba-based architecture to HSI reconstruction from snapshot compressive measurements. Sp3ctralMamba addresses key challenges in spectral recovery by integrating both frequency-domain representations and domain-specific priors into a novel joint SSM framework. The core contribution of Sp3ctralMamba lies in the design of the S3Mamba (S3MAB) block, a structured module that performs parallel scans of 3D HSI embeddings using three mechanisms: (1) vanilla scans for capturing global context across spatial dimensions; (2) local scans to emphasize fine-grained spatial details important for sparse spectral regions; and (3) spiral frequency-domain scans to exploit inter-band relationships and enhance spectral correlation modeling. Experimental results demonstrate that Sp3ctralMamba achieves SoTA performance, outperforming prior CNN-, Transformer-, and MLP-based methods in both PSNR and SSIM metrics.

Table 3 provides a comprehensive comparison of the representative CNN-, Transformer-, MLP- and Mamba-based categories of the E2E learning methods for HS image reconstruction. Each category is analyzed in terms of its core architectural components, representative models, and design characteristics relevant to HS image reconstruction. The comparison highlights the trade-offs among these approaches, particularly in their ability to capture spatial–spectral correlations, handle global dependencies, and balance reconstruction accuracy with computational efficiency. This classification serves to guide future research directions and practical implementation choices in developing robust and efficient reconstruction networks tailored for snapshot compressive imaging systems. Despite the notable advancements, these brute-force E2E learning approaches face inherent limitations: (1) they fail to exploit the underlying physical degradation models associated with the imaging process, and (2) they often suffer from limited interpretability, making it challenging to analyze or justify the reconstruction outcomes from a principled perspective.

### 3.3. Deep Unfolding Model (DUM)

To harness the complementary strengths of model-based and deep learning-based approaches, model-guided learning frameworks have garnered considerable attention in the field of HS image reconstruction [31,44,45]. Early efforts in this direction focused on plug-and-play (PnP) algorithms [44,45], which integrate pre-trained deep denoisers as implicit priors within an iterative optimization framework. In this setup, the denoiser functions as a regularization component that steers the reconstruction process toward plausible solutions. However, the use of fixed pre-trained denoisers in PnP frameworks often limits adaptability to the specific characteristics of HSI data, resulting in suboptimal reconstructions and slower convergence. To address these limitations, recent work has shifted toward the deep unfolding (or unrolling) paradigm [33,34,46,47,48,68], wherein the iterative steps of a model-based optimization algorithm are reformulated as a trainable deep neural network. This approach enables end-to-end learning of both the data fidelity and prior components, while explicitly incorporating known degradation models.

To revisit the optimization formulation presented in Equation (Equation 7), we introduce an auxiliary variable z to facilitate the use of the half quadratic splitting (HQS) technique. This method reformulates Equation (Equation 7) into the following unconstrained minimization problem:(9)$$\left(\hat{x},\hat{z}\right)=\underset{x,z}{\arg\;\min}\frac{1}{2}{∥y-\Phi x∥}_{2}^{2}+\eta R\left(z\right)+\frac{\mu }{2}{∥x-z∥}_{2}^{2},$$
where μ is a penalty parameter that enforces consistency between the variables x and z. By decoupling the variables x and z, Equation (Equation 9) can be efficiently solved via alternating minimization over two subproblems:(10)$$\begin{array}{cc} \; & x^{k+1}=\underset{x}{\arg\;\min}{∥y-\Phi x∥}_{2}^{2}+\mu {∥x-z^{k}∥}_{2}^{2},\\ \; & z^{k+1}=\underset{z}{\arg\;\min}\frac{\mu }{2}{∥z-x^{k+1}∥}_{2}^{2}+\eta R\left(z\right). \end{array}$$
These two subproblems are solved iteratively until convergence. The first subproblem in Equation (Equation 10), which enforces the data fidelity constraint, admits a closed-form solution and can be interpreted as an inverse projection step. This solution is given by:(11)$$x^{k+1}=z^{k}+\frac{1}{1+\mu }\Phi^{T}{\left(\Phi \Phi^{T}\right)}^{-1}\left(y-\Phi z^{k}\right).$$

The second subproblem in Equation (Equation 10), associated with the regularization term, is typically addressed using a learned image prior to implementation via a deep neural network. The update at the (k+1)-th Iteration can thus be represented as:(12)$$z^{k+1}=f_{NN}^{k+1}\left(x^{k+1}\right).$$
where $f_{NN}^{k+1}\left(\cdot \right)$ denotes a neural network parameterized to capture the structural prior of HS images, often tailored for denoising or feature enhancement. This learning-based regularization enhances both the adaptability and performance of the unfolding framework across diverse sensing configurations. These two subproblems are alternately solved in an iterative, multi-stage manner, where each stage corresponds to one iteration of the optimization process. This iterative scheme underpins many model-based deep unfolding networks, where learnable modules are embedded into each iteration to improve flexibility and reconstruction quality. The overall implementation of the DUM is illustrated in Figure 3.

The state-of-the-art (SoTA) DUMs are mainly devoted to (1) effective and efficient prior learning network design for spectral–spatial feature synergy and memory-enhanced unfolding and (2) adaptive degradation modeling in the data fidelity term. Prior to 2021, most prior learning networks were predominantly constructed using CNN architectures [32,53,54]. However, there has since been a paradigm shift towards Transformer-based models [33,34,46,47,48,68,69,70]. More recently, sequence modeling paradigms such as Mamba have emerged as promising alternatives to Transformers, owing to their state space model-based design that enables linear-time computation and long-range dependency modeling with lower memory consumption. One recent study now integrates Mamba-based modules for efficient and scalable prior learning in deep unfolding [71]. In the following, we introduce several representative deep unrolling models (DUMs) that leverage either Transformers or Mamba-based architectures as the backbone for prior learning.

In the following, we introduce several representative deep unrolling models (DUMs) that leverage Transformers as the backbone for prior learning.

**Degradation-Aware Unfolding Half-Shuffle Transformer (DAUHT) [33]:** DAUHT explicitly incorporates degradation modeling into the unfolding process by leveraging a half-shuffle Transformer module. The model alternates between enforcing data fidelity and applying a learned degradation-aware prior, thereby capturing both local and non-local spectral–spatial dependencies. The half-shuffle attention mechanism improves the efficiency of the self-attention computation while mitigating the influence of hardware-induced degradation artifacts.

**Pixel Adaptive Deep Unfolding Transformer (PADUT) [34]:** This approach aims to enhance the DUM with pixel-adaptive mechanisms within its Transformer modules. By adjusting the reconstruction parameters on a per-pixel basis, PADUT effectively accounts for spatial variability in HS data. This adaptive strategy enables the DUM to capture fine-grained local variations while simultaneously modeling global contextual information, leading to more accurate and robust reconstructions.

**Memory-Augmented deep Unfolding Network (MAUN) [46]:** MAUN augments the traditional DUM with an external memory component that stores and refines intermediate features across iterations. This memory module promotes the capture of long-range dependencies and historical reconstruction information, which is particularly beneficial for recovering subtle spectral details. The MAUN thereby achieves improved convergence and reconstruction quality through the efficient reuse of contextual information.

**Residual Degradation Learning Unfolding Framework (RDLUF) [68]:** RDLUF integrates residual learning into the deep unfolding process, directly modeling and compensating for degradation effects. By employing mixing priors that capture both spectral and spatial characteristics, the framework effectively disentangles the reconstruction problem into manageable subproblems. The residual learning component helps in iteratively refining the reconstructed signal, while the dual-prior strategy ensures that both global context and local details are maintained.

**Dual Prior Unfolding (DPU) [47]:** DPU decomposes the reconstruction process into two intertwined subproblems, each governed by distinct priors. One prior typically captures the data fidelity and degradation characteristics, while the other encodes the intrinsic image structure. The alternating minimization guided by these dual priors enables the network to robustly recover the high-dimensional HS image by enforcing consistency in both the measurement and prior domains.

**Spectral–Spatial Rectification (SSR) [48]:** SSR focuses on rectifying errors that arise in spectral snapshot compressive imaging by incorporating a spectral–spatial rectification module into the unfolding iterations. The rectification step functions to correct inconsistencies between the spectral and spatial domains, ensuring that the final reconstruction preserves both the global spectral signature and the fine local spatial details. This integrated strategy enhances overall fidelity and mitigates error accumulation inherent in highly compressed reconstructions.

**Highly Generalized Unfolding Model (HGUM) [70]:** SSR focuses on rectifying errors that arise in spectral snapshot compressive imaging by incorporating a spectral–spatial rectification module into the unfolding iterations. The rectification step functions to correct inconsistencies between the spectral and spatial domains, ensuring that the final reconstruction preserves both the global spectral signature and the fine local spatial details. This integrated strategy enhances overall fidelity and mitigates error accumulation inherent in highly compressed reconstructions.

**Mamba-Inspired Joint Unfolding Network (MiJUN) [71]:** To further enhance the accuracy and stability of HS image reconstruction in snapshot compressive imaging, the Mamba-Inspired Joint Unfolding Network (MiJUN) introduces a physics-guided deep unfolding framework augmented with structured state space modeling. MiJUN combines the interpretability and convergence benefits of deep unfolding networks (DUNs) with the sequence modeling capacity of Mamba-based architectures. By integrating a selective state space model (SSM) with an attention mechanism, this method effectively bridges the gap between Transformer-style architectures and implicit sequence modeling. Experimental results confirm MiJUN’s superior reconstruction fidelity.

In summary, these DUMs represent a significant shift toward hybrid models that combine principled optimization with learnable components. They address the limitations of purely data-driven approaches by embedding physical degradation models and adaptive mechanisms into the reconstruction process, thereby achieving improved interpretability, convergence speed, and reconstruction quality in HS imaging applications.

## 4. Hypespectral Image Datasets

Datasets play a pivotal role in assessing the performance of HS image reconstruction algorithms. In recent years, several emerging HS datasets have become available, facilitating both the training and validation of deep learning frameworks. In the following section, we provide an overview of open-source datasets that have significantly contributed to advancing research in HS image reconstruction.

**CAVE Dataset:** The CAVE dataset [72] comprises 32 high-quality HS images of various objects and scenes captured under controlled laboratory conditions. Each image typically has a spatial resolution of 512×512 pixels and is acquired over a spectral range spanning approximately, sampled in 400–700 nm about 31 narrow spectral bands. Owing to its high spectral fidelity and controlled acquisition conditions, the CAVE dataset has become a de facto benchmark for evaluating reconstruction algorithms under idealized scenarios.

**Harvard Dataset:** The Harvard dataset [73] comprises approximately 75 high-resolution HS images (50 under daylight, 25 under artificial/mixed illumination), and is captured under controlled illumination conditions using a Nuance FX camera with an apo-chromatic lens. With roughly 31 spectral channels spanning the visible range (approximately 420–720 nm), it is widely used for developing and testing algorithms that require precise spectral fidelity.

The Harvard dataset, frequently referenced in the spectral recovery literature, consists of a relatively small set of hyperspectral images captured in a well-calibrated environment. Characterized by high spatial and spectral resolution, the images in this dataset are obtained under controlled illumination, covering a similar visible spectral range with narrow band spacing. The dataset’s uniformity and precision make it particularly useful for assessing the performance of reconstruction methods in terms of spectral accuracy and fine detail recovery.

**ICVL Dataset:** Developed by research groups at institutions such as Imperial College London, the ICVL dataset [74] offers an extensive collection of HS images that capture natural scenes in both indoor and outdoor environments. Typically containing 201 images, each sample in the ICVL dataset is acquired across the visible spectrum (often with 31 to 33 bands) and exhibits a higher level of diversity in scene content. This large-scale dataset serves as an excellent resource for benchmarking RGB to hyperspectral reconstruction methods, as it challenges algorithms with variations in illumination, texture, and natural content.

**KAIST Dataset:** The KAIST dataset [75] includes around 30 HS images acquired from real-world outdoor environments. Typically, these images have a resolution of approximately 2704×3376 pixels and feature about 28 spectral channels. Its realistic acquisition conditions highlight the challenges encountered in practical imaging scenarios, such as noise and illumination variability. Table 4 summarizes four widely adopted HSI datasets in the community, each characterized by small to medium sizes. These datasets vary in key attributes such as data volume, spatial resolution, number of spectral channels, and scene diversity. As HS imaging technology continues to evolve, the availability of larger and more diverse datasets is anticipated, thereby further propelling the research in this field.

## 5. Result Comparison and Analysis

Recent SoTA studies frequently adopt the extended CAVE dataset for model training, owing to its moderate spatial resolution and diverse scene content, which collectively facilitate efficient learning and robust generalization. For performance evaluation, it is common practice to utilize a representative subset comprising 10 cropped HS scenes from the KAIST dataset, as demonstrated in several recent studies [23,24,33,34,42,68]. The simulation spectral snapshot imaging is widely set as the measurement process in the CASSI. To quantitatively assess reconstruction quality, two widely employed evaluation metrics are the peak signal-to-noise ratio (PSNR), which measures pixel-wise fidelity, and the structural similarity index (SSIM), which captures perceived structural consistency between the reconstructed and ground-truth images.

We conducted a thorough comparative analysis of different types of SoTA HS image reconstruction methods. In the CASSI setting, the SoTA techniques included three model-based approaches: TwIST [50], GAP-TV [21], and DeSCI [18], and seven E2E methods: λ-Net [25], DSSP [26], TSA-Net [23], HDNet [24], MST [42], CST [43] and DWMT [60], and 11 DUMs: DGSMP [32], GAP-Net [54], ADMM-Net [53], DAUHST [33], PADUT [34], MAUN [46], RDLUF-L [68], DPU [47], SSR [48], and MiJUN [71]. All methods were trained using identical training datasets and hyperparameter settings to ensure a fair comparison, and performance was evaluated under the same experimental conditions. Quantitative results were computed using two standard image quality metrics, PSNR and SSIM, across 10 simulated hyperspectral scenes. The comparative results are summarized in Table 5(a). As shown in Table 5(a), the E2E learning-based methods consistently outperform conventional model-based approaches, demonstrating the effectiveness of data-driven frameworks in complex reconstruction tasks. Notably, Transformer-based E2E models such as MST [42], CST [43], and DWMT [60] exhibit superior performance over CNN-based alternatives including λ-Net [25], DSSP [26], TSA-Net [23], and HDNet [24]. Moreover, DUMs that incorporate Transformer-based prior learning further enhance reconstruction accuracy by effectively leveraging both model-based interpretability and deep feature learning. Figure 4 illustrates the visualization comparison of 12 HS reconstruction methods by evaluating three out of the 28 spectral channels from Scene 3 including the reconstructed images and the differences with the corresponding ground-truth.

In 2023, the FPFA sensor [17] has been introduced, and consequently, only a limited number of studies such as MG-S2MLP [56], SPECAT [55] and Sp3ctralMamba [57] have investigated HS image reconstruction from snapshot measurements acquired using the FPFA setting. All three methods adopt an E2E learning paradigm, which enables the development of reconstruction models with significantly reduced parameter sizes and lower computational overhead. Table 5(b) presents the comparative results across ten simulated scenes. Owing to the effective encoding mechanism employed by the 3D sensing mask, these E2E-based methods attain PSNR values comparable to those of the best-performing DUM-based method with Transformer architecture, SSR [48], while achieving notably higher SSIM scores, despite their substantially lower computational demands.

To examine the differences in reconstruction performance between the CASSI and FPFA sensing configurations, we compute the PSNR for each spectral band of the HS images reconstructed using the SSR and MG-S2MLP methods. Figure 5 presents the average spectral-wise PSNR values obtained across 10 simulated scenes, as well as the individual results for Scene 1 and Scene 2. As shown in Figure 5, the spectral-wise PSNR distributions reveal distinct reconstruction characteristics for the HS images obtained using SSR under the CASSI setting and MG-S2MLP under the FPFA setting. Specifically, the SSR method applied to CASSI measurements tends to yield higher PSNR values at the spectral extremes (i.e., the shorter and longer wavelengths), while the performance deteriorates in the middle spectral bands. In contrast, MG-S2MLP under the FPFA configuration demonstrates significantly higher PSNR values in the central spectral bands, with much lower performance at the spectral boundaries. Furthermore, the PSNR values obtained from MG-S2MLP reveal substantially greater variability across spectral bands compared to those from SSR, indicating a less uniform reconstruction performance. These observations underscore that the characteristics of the reconstructed HS images are intrinsically linked to the underlying measurement configurations. By systematically analyzing these spectral-wise reconstruction patterns, valuable insights can be gained into the nature and distribution of information captured during the measurement process, thereby informing the design of more effective and tailored imaging systems. In addition, the spectral-wise PSNR analysis offers guidance for enhancing reconstruction algorithms, enabling adaptive calibration strategies that account for band-specific reconstruction quality.

## 6. Challenges and Trends

Previously, we presented different categories of SoTA HS image reconstruction methods and compared their quantitative performance in terms of PSNR and SSIM. This section summarizes the key insights from the surveyed methods and presents our perspective on the current challenges and future trends in HS image reconstruction. Although recent advances, particularly in deep unfolding and Transformer-based models, have made significant progress on reconstruction accuracy, critical obstacles still remain, restricting their wide applicability and robustness. We discuss three core challenges: spectral fidelity, computational efficiency, and generalization capability, and outline prospective research directions to address them.

**Spectral distortion:** One of the key challenges in HS image reconstruction is spectral distortion, i.e., errors in the reconstructed spectral signatures that misrepresent the material composition of the scene. Despite improvements in spatial fidelity (higher PSNR and SSIM), many SoTA methods especially those based on pixel-wise loss functions, fail to preserve the subtle inter-band relationships necessary for accurate spectral interpretation, and then results in large variations in PSNR scores across different bands illustrated in Figure 5, indicating lower spectral fidelity. To mitigate spectral distortion and enhance the fidelity of reconstructed HS data, future research is expected to advance in the following directions: (1) spectral-aware loss functions such as incorporating the metrics: spectral angle mapper (SAM), correlation-based losses, or perceptual spectral loss to enforce fidelity across bands; (2) spectral attention and context modeling to dynamically focus on informative spectral bands and model inter-band dependencies; (3) physics- and domain-informed learning through embedding spectral priors or physical constraints to guide reconstructions toward physically plausible outputs; (4) adaptive and band-wise modeling to flexibly handle spectral variations.

**Computational efficiency:** HS image reconstruction involves recovering high-dimensional data from compressed measurements, which brings significant computational and memory demands on reconstruction models. Deep models, especially Transformer-based ones, can become computational bottlenecks, particularly in edge devices or real-time settings. To enable practical and scalable deployment of HS image reconstruction methods, especially in real-time and resource-constrained scenarios, future research is expected to focus on several key trends: (1) lightweight and efficient network design to reduce model complexity while maintaining reconstruction quality; (2) model compression and acceleration through pruning, quantization, or knowledge distillation techniques to reduce the model size and inference latency; (3) Neural Architecture Search (NAS) for automatically discovering optimal architectures tailored to HS image reconstruction tasks; (4) hardware-aware co-design, aiming to align model architectures with edge-friendly hardware (e.g., FPGA or edge AI chips) for real-world deployment.

**Generalization capabilty:** A persistent concern in data-driven HS image reconstruction is a generalization referring to the model’s ability to perform reliably across different sensors, scenes, and acquisition conditions. Most learning-based methods perform well on curated datasets but degrade significantly on out-of-distribution or real-world measurements. Therefore, improving generalization is a critical step toward the reliable and scalable deployment of HSI reconstruction models in real-world scenarios. To enhance the robustness and adaptability of HS image reconstruction models, future research may focus on the following directions: (1) domain adaptation and generalization, including techniques such as unsupervised domain adaptation or meta-learning to adapt models to new domains without extensive retraining; (2) physics-guided hybrid models, which incorporates physical priors or signal modeling to provide inductive bias and improve generalization across datasets; (3) data diversity and augmentation, by synthesizing larger and more diverse datasets, and simulating realistic perturbations to improve resilience; (4) unsupervised and self-supervised learning, which leverages inherent structures and priors in the captured measurements to learn more generalizable representations without relying on extensive labeled data.

By addressing the aforementioned challenges through thoughtful architectural design, domain integration, and learning strategies, it is expected that future research potentially enhance the reliability, scalability, and applicability of HS image reconstruction in real-world settings.

## 7. Practical Implementations of HS Imaging Techniques

While most HS image reconstruction research focuses on algorithmic innovation and benchmark performance, the true value of these advances is realized through their deployment in real-world applications. Recent developments in spectral compressive sensing and learning-based reconstruction have made it increasingly feasible to apply HS imaging technology across diverse domains. This section introduces several potential areas, where reconstructed HSI data are being practically implemented, emphasizing both the benefits and challenges of translation from theory to practice.

**Agriculture and food safety:** HS imaging is widely used for crop monitoring and food quality control such as detecting plant stress or disease and assessing fruit ripeness or spoilage [76,77]. In practice, compressive HS images have been explored for precision agriculture (e.g., plant chlorophyll and disease mapping) and food inspection (e.g., bruise or spoilage detection) by reconstructing spectral images from compressed measurements. For instance, Bian et al. [40] developed a snapshot hyperspectral imager (HyperspecI) using a 4 × 4 broadband mosaic mask and a CNN-based decoder (SRNet) to reconstruct each frame. The captured HS images by this device are deployed to evaluate plant health (chlorophyll “SPAD” index and fruit-soluble solid content) and to detect apple bruises in real scenes. Experimental results manifest high-fidelity reconstruction of wideband spectra, enabling non-destructive measurements in real time. Jia et al. [78] explored a multiple-ROI compressive framework (HCSMAROI) for crop phenotyping with plant leaves. This method first segments leaf regions from the background and then applies block-based compressive sensing only on the plant ROIs. The recently emerging systems illustrate the potential of the compressive hardware plus learned models for real-world agri/food imaging tasks.

**Medical imaging and diagnostics:** HS imaging systems are increasingly being integrated into medical imaging applications due to their ability to capture rich spectral–spatial information rapidly. Recent advancements in reconstruction algorithms have further enhanced their applicability in clinical settings. For surgical guidance and intraoperative imaging, Ma et al. [79] investigated a dual-camera setup and produced high-resolution HS images by fusing the observed data from both cameras. The reconstruction of HS images has demonstrated improved visualization of small vessels and tumors, enhancing surgical outcomes. Lee et al. [80] leveraged the captured spectral data to assess tissue oxygenation levels, providing insights into the healing process. More recently, Huang et al. [81] employed a snapshot HS imaging technique to identify skin lesions in dermatology. By analyzing spectral signatures, clinicians can distinguish between various skin conditions, potentially leading to earlier and more accurate diagnoses.

## 8. Conclusions

In this survey, we have conducted a comprehensive and systematic review of recent advancements in HS image reconstruction from compressive measurements, particularly focusing on snapshot-based spectral compressive imaging systems. The reconstruction problem is fundamentally rooted in the mathematical modeling of the compressive sensing process, which establishes a relationship between the observed low-dimensional snapshot measurement and the high-dimensional latent HS. This formulation serves as the theoretical foundation for the design and development of diverse reconstruction algorithms. We categorize existing reconstruction methods into three main paradigms: model-based approaches, which rely on handcrafted priors and iterative optimization; end-to-end deep learning-based methods, which directly learn the mapping from measurements to reconstructed HS images; and deep unfolding-based approaches, which integrate the interpretability of model-based techniques with the learning capability of deep networks. For each category, we highlight representative methods and analyze their architectural designs, strengths, and limitations. Furthermore, we evaluate several state-of-the-art methods using two widely adopted benchmark datasets in the field. Through a comparative study of both quantitative results and qualitative reconstructions, we provide a consolidated perspective on the relative performance of each approach, offering practical guidance for selecting appropriate mechanisms in various HS image reconstruction scenarios.

Finally, we discuss key challenges that persist in the field, including spectral distortion, computational inefficiency, and limited generalization across diverse scenes and acquisition conditions. We also identify emerging research directions aimed at overcoming these limitations, such as the use of spectral-aware learning strategies, lightweight model design, domain adaptation, and physics-informed architectures. We hope that this survey serves as a valuable reference for researchers and practitioners, and inspires future innovations in hyperspectral image reconstruction.

## Figures and Tables

**Figure 1 sensors-25-03286-f001:**
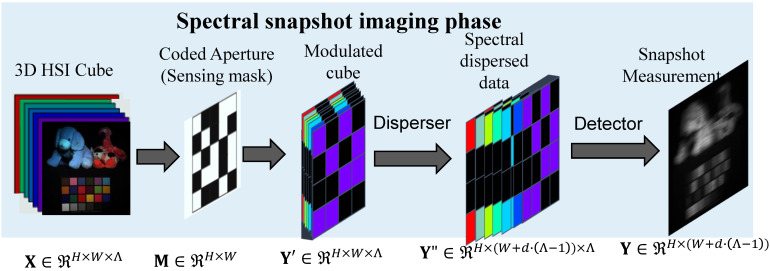
The conceptual scheme of the spectral compressive imaging process in the CASSI system.

**Figure 2 sensors-25-03286-f002:**
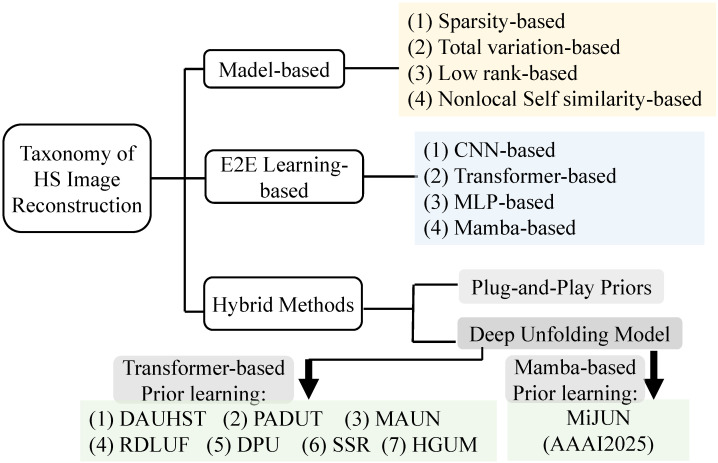
The taxonomy of the HS image reconstruction algorithms.

**Figure 3 sensors-25-03286-f003:**
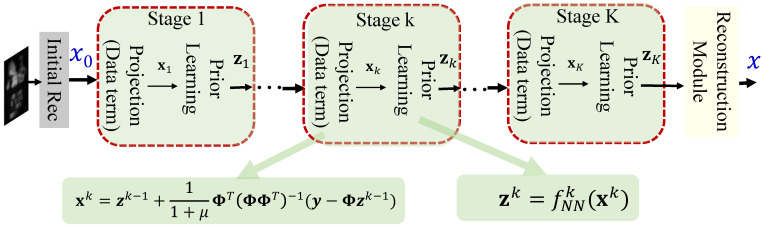
An overall implementation of the deep unfolding model.

**Figure 4 sensors-25-03286-f004:**
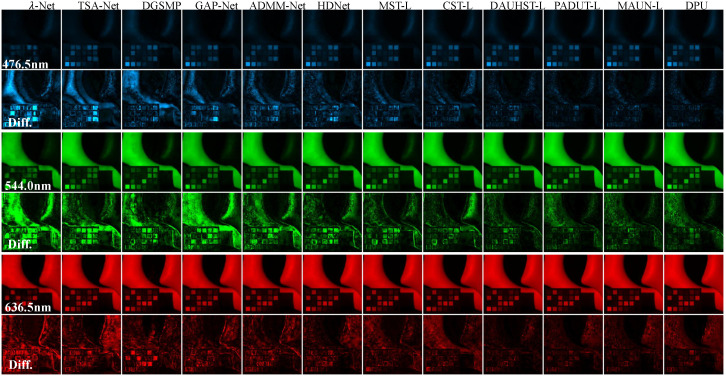
Visual quality comparison of Scene 3 (out of 28) spectral channels and the difference images between the ground-truth and reconstruction with 12 SOTA methods.

**Figure 5 sensors-25-03286-f005:**
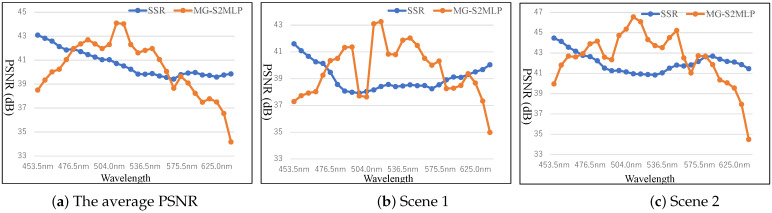
The spectral-wise PSNR distributions of the HS images reconstructed using the SSR and MG-S2MLP methods.

**Table 1 sensors-25-03286-t001:** Comparisons between the CASSI and FPFA sensors.

Aspect	CASSI	FPFA
Encoding method	Coded aperture + dispersion	Fabry–Pérot filter array
Spectral Resolution	10–20 nm	10 nm (visible range)
Light Throughput	Lower (50% loss at aperture)	High (45% average transmission)
Key Advantage	Flexibility in spectral range	Compact, high sensitivity, real-time

**Table 2 sensors-25-03286-t002:** Summary of model-based HS image reconstruction methods in spectral compressive sensing.

Prior Type	Underlying Principle	Advantages	Limitations/Challenges	Representative Works/References
Sparsity-based	Assumes that the HS data (or its transform coefficients) have a sparse representation.	Efficient recovery leveraging CS theory. Convex formulations (via $\ell_{1}$-norm) with established theoretical guarantees.	Performance depends on the chosen transform/dictionary. Does not fully exploit inherent spatial correlations. Sensitive to noise and model mismatches.	Figueiredo et al. (GPSR) [22]; Bioucas-Dias et al. (TwIST) [50].
TV-based	Utilizes total variation regularization to enforce local smoothness while preserving sharp edges by penalizing the image gradient.	Effective at denoising while maintaining significant edges. Computationally manageable via proximal methods.	May lead to staircase artifacts or over-smoothing in textured areas. Does not explicitly capture nonlocal repetitive structures.	Yuan et al. (GAP-TV) [21]
Low-Rank-based	Exploits the observation that HS images lie in a low-dimensional spectral subspace due to the high correlation among spectral bands.	Captures global spectral correlations effectively. Reduces noise and computational dimensionality.	Involves computationally expensive operations (e.g., repeated SVD, tensor decompositions). May oversmooth local details if applied globally.	Zhang et al. (Low-Rank Matrix Recovery) [41]
NSS-based	Leverages the phenomenon that similar image patches appear at different, non-adjacent locations within the image to enforce a low-rank structure on groups of similar patches.	Preserves fine structural details and textures by utilizing repetitive patterns across the image. Enhances noise suppression by enforcing consistency across nonlocal patches.	Introduces increased computational complexity due to patch matching and grouping. Performance is sensitive to parameters (e.g., patch size, similarity thresholds).	He et al. (Non-local meets global) [20]

**Table 3 sensors-25-03286-t003:** Summary of E2E learning methods for HS image reconstruction in spectral snapshot imaging systems.

Architecture Type	Representative Models/Examples	Key Architectural Components	Design Considerations and Strengths
CNN-based	TSA-Net [23], λ-Net [25], HDNet [24], NAS [30]	Convolutional layers with local receptive fields Encoder–decoder (U-shaped) structures Residual/skip connections and iterative refinement modules Simple mask incorporation (e.g., element-wise operations)	Efficient in extracting local spatial–spectral features Lower computational and memory costs Well established for image restoration tasks, though limited in modeling global dependencies
Transformer-based	MST [42], CST [43], S^2^-Tran [59], DWMT [60], SPECAT [55]	Self-attention mechanisms (e.g., spectral-wise multi-head self-attention) Dual-window/multiscale attention schemes Mask-guided modules for high-fidelity region prioritization Integration of residual connections to aid optimization	Excellent at capturing global context and long-range dependencies Can explicitly incorporate physical priors (mask information) into attention Generally higher computational and memory requirements compared to CNNs
MLP-based	MG-S2MLP [56], SSMLP [66], MLP-AMDC [67]	MLP-based token mixing for spatial and spectral dimensions Cycle Fully Connected layers (CycleFC) for efficient spatial interaction Multi-level information fusion and deep supervision modules Lightweight mask-guidance integration via MLP blocks	Lower computational overhead (linear complexity) vs. attention-based methods Effective global dependency modeling without quadratic cost Simpler architectures while capturing essential spatial–spectral interactions
State Space (Mamba-based)	Sp3ctralMamba [57]	Joint structured state space model (SSM) blocks: S3Mamba (S3MAB) Parallel spectral band scanning: vanilla, local, and spiral frequency-domain scans Hierarchical decomposition of 3D HSI embedding Integration of energy and structural priors for regularized learning	Efficient implicit attention mechanism with high-order context modeling Incorporates physical priors and frequency-domain information for improved fidelity Demonstrates superior performance on both synthetic and real-world benchmarks

**Table 4 sensors-25-03286-t004:** Summary of benchmark HS image datasets.

Dataset	# of Images	Spatial Resolution	Spectral Channels	Acquisition Conditions
CAVE	32	512 × 512	∼31 (400–700 nm)	Controlled laboratory environment
Harvard	∼50	High resolution (varies)	∼31 (420–720 nm)	Uniform, controlled illumination
ICVL	201	∼1392 × 1040	31 (approx.)	Natural scenes with diverse content
KAIST	∼30 (approx.)	2704 × 3376 (approx.)	28 (approx.)	Outdoor/real-world settings

**Table 5 sensors-25-03286-t005:** Quantitative evaluation of different SoTA methods on 10 simulation scenes. Red font indicates the best performance and blue for the second-best.

(a) For the captured measurements in CASSI setting													
**Methods**	**Params**	**GFLOPs**	**s1**	**s2**	**s3**	**s4**	**s5**	**s6**	**s7**	**s8**	**s9**	**s10**	**Avg**
TwIST [50]	-	-	25.16	23.02	21.40	30.19	21.41	20.95	22.20	21.82	22.42	22.67	23.12
			0.700	0.604	0.711	0.851	0.635	0.644	0.643	0.650	0.690	0.569	0.669
GAP-TV [21]	-	-	26.82	22.89	26.31	30.65	23.64	21.85	23.76	21.98	22.63	23.1	24.36
			0.754	0.610	0.802	0.852	0.703	0.663	0.688	0.655	0.682	0.584	0.669
DeSCI [18]	-	-	27.13	23.04	26.62	34.96	23.94	22.38	24.45	22.03	24.56	23.59	25.27
			0.748	0.620	0.818	0.897	0.706	0.683	0.743	0.673	0.732	0.587	0.721
λ-Net [25]	62.64M	117.98	30.10	28.49	27.73	37.01	26.19	28.64	26.47	26.09	27.50	27.13	28.53
			0.849	0.805	0.870	0.934	0.817	0.853	0.806	0.831	0.826	0.816	0.841
DSSP [26]	33.85M	64.42	31.48	31.09	28.96	35.56	28.53	30.83	28.71	30.09	30.43	28.78	30.35
			0.856	0.842	0.823	0.902	0.808	0.877	0.824	0.881	0.868	0.842	0.852
TSA-Net [23]	44.25M	110.06	32.03	31.00	32.25	39.19	29.39	31.44	30.32	29.35	30.01	29.59	31.46
			0.892	0.858	0.915	0.953	0.884	0.908	0.878	0.888	0.890	0.874	0.894
HDNet [24]	2.37M	154.76	35.14	35.67	36.03	42.30	32.69	34.46	33.67	32.48	34.89	32.38	34.97
			0.935	0.940	0.943	0.969	0.946	0.952	0.926	0.941	0.942	0.937	0.943
MST-L [42]	2.03M	28.15	35.40	35.87	36.51	42.27	32.77	34.80	33.66	32.67	35.39	32.50	35.18
			0.941	0.944	0.953	0.973	0.947	0.955	0.925	0.948	0.949	0.941	0.948
CST-L [43]	3.00M	40.01	35.96	36.84	38.16	42.44	33.25	35.72	34.86	34.34	36.51	33.09	36.12
			0.949	0.955	0.962	0.975	0.955	0.963	0.944	0.961	0.957	0.945	0.957
DWMT [60]	14.48M	46.71	36.46	37.75	38.47	44.23	33.99	36.17	35.22	34.56	37.41	34.99	36.82
			0.957	0.963	0.965	0.984	0.963	0.970	0.949	0.968	0.965	0.959	0.964
DGSMP [32]	3.76M	84.77	33.26	32.09	33.06	40.54	28.86	33.08	30.74	31.55	31.66	31.44	32.63
			0.915	0.898	0.925	0.964	0.882	0.937	0.886	0.923	0.911	0.925	0.917
GAP-Net [54]	4.27M	78.58	33.74	33.26	34.28	41.03	31.44	32.40	32.27	30.46	33.51	30.24	33.26
			0.911	0.900	0.929	0.967	0.919	0.925	0.902	0.905	0.915	0.895	0.917
ADMM-Net [53]	4.27M	78.58	34.12	33.62	35.04	41.15	31.82	32.54	32.42	30.74	33.75	30.68	33.58
			0.918	0.902	0.931	0.966	0.922	0.924	0.896	0.907	0.915	0.895	0.918
DAUHST-L [33]	6.15M	79.50	37.25	39.02	41.05	46.15	35.80	37.08	37.57	35.10	40.02	34.59	38.36
			0.958	0.967	0.971	0.983	0.969	0.970	0.963	0.966	0.970	0.956	0.967
PADUT-L [34]	5.38M	90.46	37.36	40.43	42.38	46.62	36.26	37.27	37.83	35.33	40.86	34.55	38.89
			0.962	0.978	0.979	0.990	0.974	0.974	0.966	0.974	0.978	0.963	0.974
MAUN-L [46]	3.77M	143.83	37.78	40.53	41.88	46.85	36.74	37.78	37.44	36.05	40.54	34.90	39.05
			0.963	0.976	0.973	0.986	0.973	0.974	0.961	0.971	0.973	0.962	0.971
RDLUF [68]	1.81M	115.16	37.94	40.95	43.25	47.83	37.11	37.47	38.58	35.50	41.83	35.23	39.57
			0.966	0.977	0.979	0.990	0.976	0.975	0.969	0.970	0.978	0.962	0.974
DPU [47]	2.85M	49.26	38.79	41.78	43.80	47.69	37.96	38.48	39.00	36.81	42.65	36.28	40.33
			0.971	0.983	0.983	0.993	0.981	0.981	0.973	0.979	0.984	0.974	0.980
SSR [48]	5.18M	78.93	38.95	41.83	44.16	48.09	38.53	38.40	39.03	38.88	42.88	36.00	40.47
			0.973	0.984	0.983	0.994	0.983	0.981	0.974	0.980	0.985	0.973	0.981
MiJUN [71]	0.56	73.67	39.26	41.78	44.31	48.53	39.30	38.22	41.00	36.72	43.84	35.56	40.86
			0.973	0.983	0.983	0.994	0.985	0.979	0.983	0.978	0.985	0.967	0.982
**(b)** For the captured measurements in FPFA setting													
**Methods**	**Params**	**GFLOPs**	**s1**	**s2**	**s3**	**s4**	**s5**	**s6**	**s7**	**s8**	**s9**	**s10**	**Avg**
MG-S2MLP [56]	0.31	15.2	39.47	42.26	41.39	45.08	39.15	39.86	38.97	37.05	40.93	37.05	40.12
			0.982	0.989	0.982	0.990	0.988	0.988	0.976	0.980	0.987	0.988	0.985
SPECAT [55]	0.29	12.4	40.24	42.40	41.43	44.90	39.62	39.90	39.41	37.49	40.45	37.90	40.39
			0.982	0.986	0.978	0.982	0.987	0.984	0.977	0.977	0.982	0.983	0.982
Sp3ctralMamba [57]	0.45	64.65	40.66	43.22	42.17	45.64	40.75	41.70	39.88	37.94	41.43	38.71	41.21
			0.989	0.992	0.988	0.990	0.993	0.991	0.986	0.988	0.986	0.989	0.989

## Data Availability

Data available in a publicly accessible repositor. The introduce datasets include CAVE, Harvar, ICVL and KAIST datasets. The CAVE dataset is available at https://cave.cs.columbia.edu/repository/Multispectral (accessed on 10 January 2025), the Harvar dataset is available at https://vision.seas.harvard.edu/hyperspec/download.html (accessed on 10 January 2025), the ICVL dataset is available at https://icvl.cs.bgu.ac.il/pages/researches/hyperspectral-imaging.html (accessed on 10 January, 2025) and the KAIST dataset is available at https://vclab.kaist.ac.kr/siggraphasia2017p1/kaistdataset.html (accessed on 10 January, 2025).

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
