# Peer review of "Recent Advancements in Hyperspectral Image Reconstruction from a Compressive Measurement"

_sensors, 2025, doi:10.3390/s25113286_

Round 1
Reviewer 1 Report
Comments and Suggestions for Authors
The authors tried to survey recent hyperspectral image reconstruction methods from a compressive measurement. Though the manuscript looks engaging, it needs to be rewritten and scientifically organised to be acceptable. Below are some of the suuggestions to improve the manuscript:
- Authors should rewrite abstract.
- Authors need to compare their work with some recently published review papers (https://doi.org/10.1007/s10462-024-11090-w, https://doi.org/10.1038/s41598-022-16223-1 etc) on hyperspectral reconstruction and discuss the novelty of their manuscript clearly.
- It is recommended to reduce use of dashes throughout manuscript.
- A summary diagram/graphical abstract can improve the clarity of the overall manuscript.
- A diagram is needed to demonstrate the architecture of the methods.
- Authors need to describe more HS reconstruction methods.
- The challenges and trends section needs to be fully rewritten. It should contain authors' view on the recent and future perspectives that can be helpful for researchers on this field.
- This manuscript lacks the discussions on practical implementation of HS reconstructions. Many recent studies used HS reconstruction in agriculture, biology, medical imaging, and so on. Authors need to discuss on these practical implementations.
Author Response
Please refer to the attached response file.

Reviewer 2 Report
Comments and Suggestions for Authors
This paper provides a comprehensive review of the techniques for hyperspectral image reconstruction from a compressive measurement, which are categorized into three groups: model-based, learning-based, and hybrid approaches. The structure of the paper is good, the mathematical assumptions are correct, and the paper provides a reasonable review. I congratulate the authors on their effort, and here are some simple comments that may help:
1. There are minor typos; please revise them thoroughly. (e.g., line 341. the numbering of references contains '?'; and line 719. 'inHS' must be 'in HS' ).
2. The mathematical notations must be improved, and please use standard notations.
3. While the interpretable deep learning frameworks are exciting, I suggest that the author improve the details of Figure 2. For example, the authors can make a suitable derivation of optimization problem 9 and then reflect this derivation in Figure 2 as a deep unfolding network.
4. Please provide the web hyperlinks of the hyperspectral image datasets stated in this paper.
5. The best and second-best obtained results in Table 5 should be highlighted, seeking clarity.
Author Response

(The authors gave the same response as above.)

Round 2
Reviewer 2 Report
Comments and Suggestions for Authors
I recommend the paper for acceptance, while there is an issue in listing the references (e.g., line 521) that the authors must carefully review before submitting the final version.